# 4-Hydroxynonenal from Mitochondrial and Dietary Sources Causes Lysosomal Cell Death for Lifestyle-Related Diseases

**DOI:** 10.3390/nu16234171

**Published:** 2024-11-30

**Authors:** Tetsumori Yamashima

**Affiliations:** Department of Psychiatry and Behavioral Science, Kanazawa University Graduate School of Medical Sciences, Takara-machi 13-1, Kanazawa 920-8040, Japan; yamashima215@gmail.com; Tel.: +81-(90)-2129-1429; Fax: +81-(76)-247-1338

**Keywords:** calpain–cathepsin hypothesis, cardiolipin, GPR40, Hsp70.1, lysosomal rupture, POMC neuron, ROS

## Abstract

Excessive consumption of vegetable oils such as soybean and canolla oils containing ω-6 polyunsaturated fatty acids is considered one of the most important epidemiological factors leading to the progression of lifestyle-related diseases. However, the underlying mechanism of vegetable-oil-induced organ damage is incompletely elucidated. Since proopiomelanocortin (POMC) neurons in the hypothalamus are related to the control of appetite and energy expenditure, their cell degeneration/death is crucial for the occurrence of obesity. In patients with metabolic syndrome, saturated fatty acids, especially palmitate, are used as an energy source. Since abundant reactive oxygen species are produced during β-oxidation of the palmitate in mitochondria, an increased amount of 4-hydroxy-2-nonenal (4-HNE) is endogenously generated from linoleic acids constituting cardiolipin of the inner membranes. Further, due to the daily intake of deep-fried foods and/or high-fat diets cooked using vegetable oils, exogenous 4-HNE being generated via lipid peroxidation during heating is incorporated into the blood. By binding with atheromatous and/or senile plaques, 4-HNE inactivates proteins via forming hybrid covalent chemical addition compounds and causes cellular dysfunction and tissue damage by the specific oxidation carbonylation. 4-HNE overstimulates G-protein-coupled receptors to induce abnormal Ca^2+^ mobilization and µ-calpain activation. This endogenous and exogenous 4-HNE synergically causes POMC neuronal degeneration/death and obesity. Then, the resultant metabolic disorder facilitates degeneration/death of hippocampal neurons, pancreatic β-cells, and hepatocytes. Hsp70.1 is a molecular chaperone which is crucial for both protein quality control and the stabilization of lysosomal limiting membranes. Focusing on the monkey hippocampus after ischemia, previously we formulated the ‘calpain–cathepsin hypothesis’, i.e., that calpain-mediated cleavage of carbonylated Hsp70.1 is a trigger of programmed neuronal death. This review aims to report that in diverse organs, lysosomal cell degeneration/death occurs via the calpain–cathepsin cascade after the consecutive injections of synthetic 4-HNE in monkeys. Presumably, 4-HNE is a root substance of lysosomal cell death for lifestyle-related diseases.

## 1. Introduction

Currently, Alzheimer’s disease, type 2 diabetes, nonalcoholic steatohepatitis (NASH), etc., constitute prevalent lifestyle-related diseases worldwide, but the causal relation among these multifactorial diseases remains poorly understood at the molecular level. Since the pathophysiology of each disease is complex and multi-faceted, very few effective treatment strategies exist to overcome them. The intracellular milieu of the brain, pancreas, and liver is rich in oxygen, glucose, and fatty acids which are necessary for ATP synthesis. However, the respiratory chain complex in the mitochondria produces reactive oxygen species (ROS) as byproducts during ATP synthesis via acetyl-CoA derived from glucose and/or fatty acids. ROS, including the hydroxyl radical (OH•) and superoxide anion (O_2_^•–^), and their precursor, hydrogen peroxide (H_2_O_2_), are abundantly produced in the mitochondria of the brain, pancreas, and liver in people with hyperphagia, obesity, or metabolic syndrome. Although hydrogen peroxide is not very reactive, its interaction with Fe^2+^ produces a very active hydroxyl radical via the Fenton reaction in mitochondria.

Free radicals play an important role in the maintenance of a homeostatic environment for cell survival and adaptation, because physiological levels of ROS are useful for many cellular activities, including gene transcription, signaling transduction, and immune response [1]. For example, hydrogen peroxide and the superoxide anion are involved in developmental signaling transduction in pancreatic β-cells, having the ability to control insulin secretion. Under physiological conditions, the concentration of ROS is subtly regulated by antioxidants. In contrast, overproduction of ROS makes neurons, β-cells, and hepatocytes with their weak antioxidant capacity vulnerable to sustained oxidative stress [2,3,4,5,6,7,8]. Excessive oxidative stress decreases the antioxidant status of cells/organs by reducing activities of reductants and antioxidative enzymes.

An imbalance in the production and neutralization of ROS causes an accumulation of ROS intermediate products which may induce sustained oxidative stress. This adversely affects the structure of cell membranes, lipids, proteins, and DNA, playing a critical role in the pathogenesis of chronic degenerative diseases. The excessive oxidation of proteins and lipids producing nitrotyrosine, aldehyde, carbonyl, etc., might decrease the activity of enzymes and growth factors by causing cellular dysfunction [5,8,9,10]. Further, peroxidative damages on the membranous phospholipids cause activation of sphingomyelinase and ceramide release. In addition, long-standing ROS react with nucleic acids by attacking the nitrogenous bases and the sugar phosphate backbone, thus inducing single- and double-stranded DNA breaks. These, combined, have been thought to cause cell degeneration/death in diverse organs and lead to different life-threatening pathological conditions, depending on the organ affected [11]. The transfer of abundant electrons (e^−^) from NADH and FADH_2_ being produced at the TCA cycle in the mitochondrial matrix to the respiratory chain in the inner membranes facilitates ROS production in all kinds of cells during stresses [12]. When the cells are exposed to oxidative stress, the cytochrome P450 biotransformation activities can generate free radicals [13]. All of these lead to insulin resistance, glucose intolerance, and liver steatosis. Based on alterations in the different organs, Hajam et al. [14] explained the pathogenesis of ROS-mediated cell degeneration/death in the representative lifestyle-related diseases. However, the interaction of each disease still remains unelucidated.

An abundance of ω-6 polyunsaturated fatty acids (PUFA) such as linoleic acids is essential in order to maintain membrane fluidity and the osmotic stability of the dynamic organelle, mitochondria. However, cardiolipin, constituting 25% of the mitochondrial inner membranes and containing four acyl groups of linoleic acids, is vulnerable to ROS produced at the electron transport chain [15,16]. Although the ROS-mediated molecular cascades are distinct by disease, the damage of mitochondria appears common to all diseases, because ROS are mainly produced in mitochondria. Accordingly, the question emerges of whether there is a common molecular cascade and/or root substance among various lifestyle-related diseases which are related to mitochondrial disorder. To identify such a common cascade or substance, if present, we suggest focusing on ROS-induced disorders of biomembranes in mitochondria. ROS-induced oxidative damage of mitochondria exerts a crucial role for developing pathologies in the different organs.

The characteristics of lifestyle-related diseases include the progressive loss of function due to cell degeneration/death in the corresponding tissues and organs. Elucidating the common molecular cascade and/or the root substance of cell degeneration/death that originates in mitochondria will contribute to the development of a novel therapeutic agent and the overall well-being of humans. In the United States, for example, there has been a 1000-fold increase in the consumption of ω-6 dietary oils during the 20th century, and this might be one of the reasons explaining the prevalence of obesity, metabolic syndrome, type 2 diabetes, NASH, Alzheimer’s disease, atherosclerosis, cardiomyopathy, and heart failure [17,18,19]. Therefore, it is reasonable to speculate that their main component, linoleic acid, and/or their peroxidation product might have a close relation to the occurrence of pathologic conditions.

This review aims to elucidate the common root substance of the three representative lifestyle-related diseases by focusing on the peroxidation product of ω-6 PUFA, especially linoleic acids, which are involved in both vegetable oils and mitochondrial inner membranes. If the root substance causing the representative diseases is a linoleate-derived peroxidation product, the development of novel agents with an improved safety and effectiveness would be possible by attenuating its toxicity.

## 2. ROS-Induced Peroxidation of ω-6 PUFA, Especially Linoleic Acid

High-fat diets rich in animal fat and American fast foods rich in vegetable oils have increased greatly around the world for the past half-century. Chronic consumption of animal fat and vegetable oils is nowadays considered one of the most crucial environmental factors leading to the prevalence of obesity and comorbid diseases. Animal fat is rich in saturated fatty acids, while vegetable oils are rich in ω-6 PUFA. Whereas most studies about diet-induced obesity focus on the role of saturated fats involved in animal fat, a growing body of evidence suggests that linoleic acids involved in vegetable oils such as soybean, rapeseed (canola), corn, and sunflower oils also contribute to the obesity epidemic.

By producing ROS, high-fat diets exert a crucial role in the development of cell degeneration/death. Saturated fatty acids like palmitate (C16:0) and stearate (C18:0) are associated with an increase in the ROS level, because β-oxidation of these fatty acids, if in excess, induces stress in the mitochondrial electron transport chain (Figure 1a and Figure 2a) [15,16]. It is recognized that chronic intake of saturated fatty acids causes metabolic alterations via inflammation not only in the brain but also in the peripheral organs. Although the hypothalamus controls the balance between energy homeostasis and obesity, the specific mechanisms regulating that balance remain elusive. Moraes et al. [20] reported that high-fat diets induce apoptosis of hypothalamic neurons, especially in the arcuate nucleus. They suggested that high-fat diets blunt leptin and insulin anorexigenic signaling in the hypothalamus and activate pro-inflammatory pathways, endoplasmic reticulum stress markers, and apoptotic signaling pathways.

The hypothalamus is a complex brain network that is responsible for sensing nutritional status and executing behavioral and metabolic responses to changes in fuel availability. It produces intrinsic peptides and neurotransmitters that influence food intake, energy balance, and glucose homeostasis. High-fat diets disrupt energy balance, cause changes in body weight regulation, and also lead to an increased hypothalamic expression of both the inflammatory cytokines (IL-1β, IL-6, and TNF-α) and proteins (SOCS3, JNK, and IKK) that are involved in inflammatory signal transduction. Whenever active, signal transduction through Toll-like receptor (TLR) 4 induces cytokines [21], and loss-of-function mutation in TLR4 prevents diet-induced obesity and insulin resistance [22]. On the contrary, the presence of an intact TLR4 protected hypothalamic neurons from high-fat-diet-induced inflammation [20]. Saturated fatty acids, unlike unsaturated fatty acids, were also proposed as triggers of the NLR family pyrin domain containing 3 (NLRP3) inflammasome, a molecular platform mediating the processing of interleukin-1β (IL-1β) in response to stress conditions [23]. TLR1, TLR2, and TLR4 enhance ROS production by recruiting mitochondria to macrophage phagosomes [24]. Much research has focused on TLR4 and NLRP3 inflammasomes, which are required for the processing and release of a key player, IL-1β. During the priming stage, activation of TLR4 upregulates NLRP3 and pro-IL-1β expression through nuclear factor-κB (NF-κB) activation [25].

In the hypothalamus, both TLR4 and NLRP3 exert dual roles, participating in the balance between inflammation via activating pro-inflammatory cytokines and cell survival via restraining further damage by controlling apoptotic pathways. However, the role of neuroinflammation in hypothalamic neuronal degeneration/death still remains to be elucidated; it is unknown why neurons, especially hypothalamic POMC neurons, chronically exposed to excessive saturated fatty acids or ω-6 PUFA develop cell degeneration/death via decreased mitochondrial dynamics and bioenergetic capacity. Understanding the exact mechanism which leads to metabolic alterations in POMC neurons by excessive and/or oxidized fatty acids may help us develop strategies for the prevention and treatment of obesity and related lifestyle-related diseases. It is also important to determine the role of ROS in dietary oil toxicity for inducing POMC neuronal degeneration/death and obesity.

Exposure to palmitic acid impairs mitochondrial function by the decrease in membrane potential and ATP production as well as the increase in β-oxidation, and ROS levels (Figure 1a and Figure 2a) [26,27]. But how do lipid-overloaded mitochondria induce stress and insulin resistance? It has been proposed that chronic intake of excessive fatty acids leads to persistent pressure on the electron transport chain of mitochondria, resulting in disruption of the redox balance and ROS-signaling (Figure 1a). Further, ROS-induced oxidation of ω-6 PUFA had been extensively studied for more than half a century. In 1980, a highly reactive α,β-unsaturated aldehyde, 4-hydroxy-2-nonenal (4-HNE) was discovered in vivo as a cytotoxic product originating from the peroxidation of liver microsomal lipids (Figure 2b,c) [28], and ω-6 PUFA, including linoleic acid (C18:2), arachidonic acid (C20:4), and γ-linolenic acid (C18:3), are precursors of 4-HNE.

**Figure 1 nutrients-16-04171-f001:**
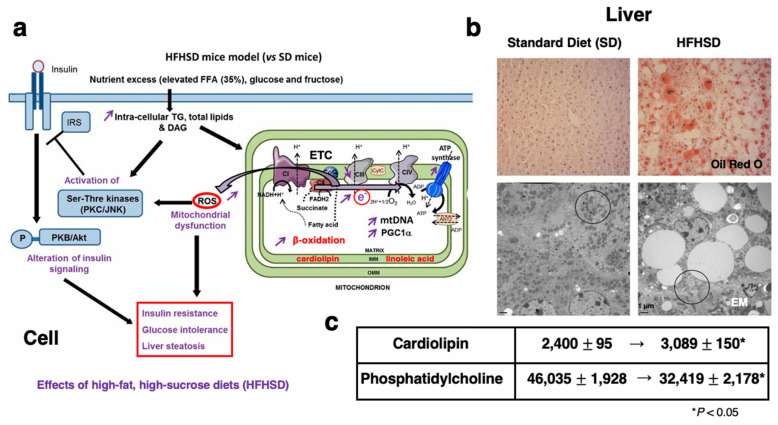
Effects of high-fat, high-sucrose diets (HFHSD) upon the mitochondria of hepatocytes as compared to standard diets (SD). (**a**) HFHSD mice show an increase in intracellular lipids such as triglyceride (TG) and diacylglycerol (DAG), which alters mitochondrial function and inhibits insulin signaling. Mitochondria are a major source of ROS such as superoxide and hydrogen peroxide, being produced at the electron transport chain (ETC). IMM: inner membrane of mitochondria, OMM: outer membrane of mitochondria. (**b**) HFHSD mice show an increase in lipid droplets in the liver (Oil Red O staining, original magnification ×200), whereas mitochondria in the hepatocytes are decreased (circles, EM; electron microscopy, bar = 1 µm). Vacuoles are consistent with lipid droplets (orange) as seen by Oil Red O staining. (**c**) In HFHSD mice, cardiolipin is significantly increased, whereas phosphatidylcholine is decreased in mitochondria in the relative quantitation by high-performance liquid chromatography. An increase in cardiolipin indicates the amplification of mitochondrial respiration by overfeeding. (Adapted from Ref. [16]; Vial et al., 2015).

**Figure 2 nutrients-16-04171-f002:**
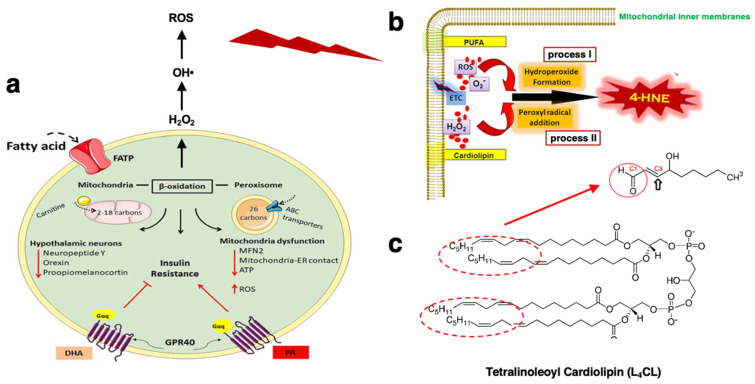
Neuronal β-oxidation (**a**) and ROS-dependent 4-HNE generation from mitochondrial cardiolipin (**b**,**c**). (**a**) Fatty acids 2–18 carbon atoms in length are transported into the mitochondria by carnitine. These fatty acids break down via β-oxidation to acetyl-CoA which is fed directly into the TCA cycle for ATP synthesis within the mitochondrial matrix. β-oxidation of palmitic acids (PAs) in neurons is associated with the reductions in the mitochondrial fusion protein MFN2, membrane potential, and ATP production as well as increases in reactive oxygen species (ROS) via the byproducts NADH and FADH_2_ (Figure 1a). In POMC neurons, PAs activate GPR40 signaling with resultant insulin resistance, whereas DHA-dependent activation of GPR40 improves insulin signaling (cited from Ref. [27]; Sánchez-Alegría and Arias, 2023). DHA, docosahexaenoic acid; PA, palmitic acid. (**b**) As a byproduct of oxidative phosphorylation at the inner membranes, mitochondria continue to produce ROS during cellular stress. Lipid peroxidation of the acyl chains of PUFA occurs via the superoxide (O_2_^−^) formed at the ETC (process I). Oxidation of the mitochondria-specific tetralinoleoyl cardiolipin (L_4_CL) by cytochrome c and H_2_O_2_ results in the addition of peroxyl radicals (process II). (**c**) Since 25% of the mitochondrial inner membranes is made up of cardiolipin containing four acyl groups, and the distribution of linoleate in the cardiolipin is 85–90%, the ETC-derived free radicals generate endogenous 4-HNE by oxidizing the inner-membrane-specific phospholipid L_4_CL. Dot circle portions in L_4_CL generate 4-HNE via oxidative injury. The highly reactive thiol or amino compounds on the C3 of the C2=C3 double bond of 4-HNE (**b**, open arrow) combine with the side chains of amino acids to form protein adducts and contribute to protein cross-linking via the Michael addition. Then, by binding the C1 carbonyl group (**b**, red circle) with primary amines such as cysteine, histidine, lysine, and arginine via forming Schiff bases, HNE induces a carbonyl injury to the substrate proteins like Hsp70.1 and BHMT (**b**,**c**: Cited from Ref. [13]; Dalleau et al., 2013).

The lipid-peroxidation product, 4-HNE, contributes to protein cross-linking (protein adducts) and induces carbonyl stress (protein carbonylation) [13]. 4-HNE possesses three reactive functions: a C2=C3 double bond, a C1=O carbonyl (aldehyde) group (Figure 2b, red circle), and a hydroxyl group at C4 (Figure 2b). 4-HNE reacts with proteins containing cysteine, histidine, lysine, and arginine residues, with lipids containing an amino group, and with nucleic acids mostly with the guanosine moiety of DNA. The reactivity relies upon both the Michael addition of thiol or amino compounds on the C3 of the C2=C3 double bond and the formation of Schiff bases between the C1 carbonyl group and primary amines [29,30,31,32,33,34]. Because of its double reactivity, after protein cross-linking via the Michael addiction, 4-HNE can induce carbonyl stress to the primary amines in the substrate protein via forming Schiff bases [13]. As 4-HNE is a highly diffusible molecule, it can spread beyond its initial production site as a paracrine signal molecule. 4-HNE exhibits a variety of biological activities, including the specific oxidation (carbonylation) of proteins, inactivation of enzymes, and mutation of DNA. In addition, probably by damaging mitochondria, the release of mtDNA to the cytosol, and the triggering of the innate immune response, 4-HNE induces expression and synthesis of interleukin-8 (IL-8), IL-1β, and tumor necrosis factor-α (TNF-α), and upregulates matrix metalloproteinase-9 (MMP-9) via the TLR4/NF-κB-dependent pathway [35].

Under physiological conditions, the cellular concentration of 4-HNE ranges from 0.1 to 3 μM. However, the concentration of 4-HNE in cells during stress conditions rises to higher concentrations of 10 μM to even 5 mM levels [29,36,37]. For example, in rats exposed to carbon tetrachloride (CCl_4_), the level of 4-HNE reaches up to 100 μM in hepatocytes [38,39,40,41]. At concentrations higher than 10 μM, 4-HNE strongly reacts with proteins, leading to their dysfunction, tissue toxicity, and ultimately organ damage. In patients with Alzheimer’s disease, for example, the blood 4-HNE levels are 3~5-fold higher, as compared to age-matched healthy control subjects. Using the method of Esterbauer and Cheeseman [42], serum 4-HNE levels are 6.0–25.2 μmol/L (median 20.6 μmol/L) in Alzheimer’s patients, and 3.3–14.5 μmol/L (median 7.8 μmol/L) in the control subjects [43]. Similarly, increased levels of 4-HNE were reported in patients with type 2 diabetes [44,45]. In patients with NASH, also, a significant increase in the serum 4-HNE levels was demonstrated, compared to those with simple steatosis [46]. Although both the extent of cellular toxicity and dysfunction, and how 4-HNE affects the cells, depend on the type and intensity of cell and stress conditions, it is probable that 4-HNE-triggers the molecular cascade which is common to Alzheimer’s disease, type 2 diabetes, and NASH [8,47,48].

Excessive endogenous linoleic-acid- and/or palmitic-acid-induced ROS generate 4-HNE from the cardiolipin of mitochondrial inner membranes (Figure 2b,c). This would explain why 4-HNE levels are much higher within cells and tissues compared to serum levels. Exogenous 4-HNE accumulates in the serum via consumption of deep-fried foods, whereas endogenous 4-HNE is generated within cells in response to stresses and can reach levels higher than those of a food source. In lifestyle-related diseases, however, long-standing circumferential stresses and daily consumption of ω-6-PUFA-containing foods would synergically elevate 4-HNE levels and exert cell and tissue damage. Intriguingly, after consecutive (5 mg/week × 6 months) injections of synthetic 4-HNE in monkeys in which the serum concentration reached that observed in humans in their 60s, significant brain, pancreas, and liver damage was observed, as shown below [48]. In the following section, we discuss the role of 4-HNE in the development of lysosome-induced cell death of neurons, β-cells, and hepatocytes.

## 3. Palmitic Acid Induces POMC Neuronal Degeneration/Death: How?

Fatty acids are made up of a hydrocarbon chain with a terminal carboxyl group. Saturated fatty acids are composed of a single aliphatic chain bond, whereas mono- or poly-unsaturated fatty acids are composed of one- or two-carbon double bonds, respectively. According to chain length, unsaturated fatty acids are divided into short-chain (2–4 carbon atoms in length), medium-chain (6–12 carbon atoms), and long-chain (14–18 carbon atoms). Short-chain unsaturated fatty acids are immediately available as an energy source, medium-chain fatty acids act as growth factors, and long-chain fatty acids make cellular membranes. GPR40 (also known as free fatty acid receptor 1: FFAR1), a member of the G-protein-coupled receptor (GPCR) family, was first described in 2003 as a receptor for long-chain fatty acids independently by two groups [49,50]. For example, unsaturated fatty acids such as docosahexaenoic acid (DHA) bind to GPR40 to synthesize BDNF and promote adult neurogenesis in the hippocampus [51,52,53,54,55] and the hypothalamus [56]. Fatty acids are mostly (95%) bound to albumin in the blood, cross the plasma membrane through fatty acid transporters (FATP) (Figure 2a) [57], and then are trafficked into specific intracellular domains or membranes for further utilization by the fatty acid binding protein (FABP) [53].

Free fatty acids are only minimally used for ATP production in the brain; only 20% of the total energy requirement in the adult brain is obtained from the oxidation of fatty acids [58], and most of it occurs in astrocytes, not in neurons. In contrast, saturated fatty acids can be metabolized through β-oxidation in both neurons and astrocytes. Most importantly, long-chain saturated fatty acids such as palmitic and stearic acids exert a role as a major energy source in metabolic alterations [27]. Obese individuals with metabolic syndrome uptake more palmitic acid in the brain as compared to lean ones [59]. Neurons can metabolize palmitic acids within mitochondria, depending on the chain length and energy requirements (Figure 2a) [60,61,62,63]. Therefore, neurons of obese individuals are subjected to higher ROS that are generated by β-oxidation of saturated fatty acids. This is observed, for example, in proopiomelanocortin (POMC) neurons which express high levels of GPR40 and FABP [51,52,54,55,56].

The arcuate nucleus, which is adjacent to the floor of the third ventricle in the mediobasal hypothalamus, contains two types of neurons that exert potent effects on food intake, energy expenditure, and glucose homeostasis (Figure 3a, light-blue area) [64]. One type is agouti-related peptide (Agrp)/neuropeptide Y (Npy) neurons which stimulate food intake and reduce energy expenditure, and thereby promote body weight gain in response to ghrelin [65,66,67]. The other type is POMC/cocaine- and amphetamine-regulated transcript (Cart) neurons which inhibit food intake and promote body weight loss via the inputs not only from leptin or insulin but also from free (non-esterified) fatty acids in the blood [68,69,70]. In POMC neurons, leptin and insulin signals are mediated by leptin receptors (Lepr), whereas fatty acid signals are mediated by GPR40 (Figure 3a). However, sustained, excessive stimulation of the GPR40 in POMC neurons triggers cell degeneration/death due to increased Ca^2+^ and μ-calpain activation [20,48,71,72].

POMC neuronal loss facilitates hyperphagia and body weight gain that lead to insulin resistance and glucose intolerance. Rodent models of obesity induced by high-fat diets showed upregulation of Hsp72 (compatible with the Hsp70.1 in humans) in POMC neurons of the arcuate nucleus (Figure 4a–d). Hypothalamic proinflammatory cytokine and NF-κB pathway genes were evident within 1 to 3 days after the intake of high-fat diets in both rats and mice. In addition, both reactive gliosis and inflammatory markers suggestive of neuronal injury were evident in the arcuate nucleus at the first week of high-fat diet feeding. Although these responses temporarily subsided within 1 to 3 weeks, POMC neurons of a high-fat mouse show disruption of mitochondrial membrane integrity and an increase in autophagosomes. At 8 months after the onset of the high-fat diet (HFD), a reduction of approximately 25% in the number of POMC cells was seen in the arcuate nucleus of mice as compared to the controls fed normal diets (Chow) over the same time period (Figure 4e,f). Hypothalamic mRNA expression of Hsp72 increases within 3 days of high-fat-diet exposure, and Hsp72 immunostaining increases substantially at 7 days in the POMC neurons of rodents (Figure 4a,b) [72]. Based on these data, Thaler et al. suggested that the transient decrease in hypothalamic inflammatory signaling was due to the rapid induction of the molecular chaperone Hsp72 (Figure 4c,d). However, whether Hsp72 upregulation is beneficial or detrimental for the occurrence of POMC neurodegeneration was not determined in that study. As discussed later, 4-HNE results in carbonylation-induced cleavage of Hsp70.1 (Figure 4g) [5,8,48], and at the same time, 4-HNE also induces Hsp70.1 upregulation through the nuclear export of Daxx (death association protein 6) (Figure 4h) [75]. HNE-induced translocation of Daxx from the nucleus to the cytoplasm releases heat shock transcription factor 1 (HSF1) and allows it bind to its DNA recognition elements to increase Hsp70.1 expression [76].

## 4. Generation of 4-HNE from Mitochondrial Cardiolipin

Palmitic acid is a 16-carbon long-chain saturated fatty acid and the most abundant (65%) saturated fatty acid in the human body. An increase in circulating palmitic acid is associated with the progression of lifestyle-related diseases, such as type 2 diabetes, cardiovascular diseases, and dementia [77,78,79]. However, the underlying cause of palmitic acid toxicity is incompletely elucidated. Palmitic acid contribution to the biosynthesis of ceramides [80,81,82] may mediate the adverse effects of high-fat diets on neurons. Ceramides are signaling molecules involved not only in neuronal development but also cellular senescence and death. In particular, ceramide 16 is involved in apoptosis [83], and ceramides 24 and 16 participate in the development of insulin resistance [84]. Although a variety of other potential mechanisms for palmitic acid in human pathology have been explored, the underlying molecular cascade responsible for dietary-fat-induced neuronal dysfunction, degeneration, and death had remained elusive except for the implication of ceramides.

As discussed, ATP generation in mitochondria also generates ROS (Figure 1a and Figure 2a,b), and the ROS enhance oxidation of carbon–carbon double bonds of ω-6 PUFA circulating in the blood and/or within biomembranes and generate endogenous 4-HNE (Figure 2b,c). ROS such as the superoxide anion and hydroxyl radicals (Figure 2b) have a very short half-life [13], whereas 4-HNE reacts with proteins nearby, becoming more stable. 4-HNE-protein adducts mediate oxidation (carbonylation) injury and protein dysfunction, and contribute to protein aggregation, e.g., in aggregates of amyloid β in the senile plaques and within atheromatous plaques [35,36,85,86,87,88]. However, where and how 4-HNE is generated in the human body has been incompletely understood [34,89].

Lipid peroxidation and subsequent 4-HNE generation in mitochondria cause major mitochondrial dysfunction [33], and thus produce more ROS and more damage in particular, via cardiolipin oxidation. Cardiolipin is uniquely localized to the mitochondrial inner membrane, contributing to the unique structure of the cristae and for the activity of the cytochrome bc1 complex (CIII) (Figure 1a) [90]. (The name ‘cardiolipin’ is derived from the fact that it was originally found in hearts, an organ particularly enriched with mitochondria.) In most mammalian tissues, the predominant form of cardiolipin is tetralinoleoyl cardiolipin (L_4_CL), and the distribution of linoleate in mitochondrial cardiolipin is around 85–90% [91,92,93]. Cardiolipin has a unique structure; it comprises two phosphate residues and four fatty acyl chains (Figure 2c). Overfeeding mice with cardiolipin results in increases in the mitochondrial inner membranes (Figure 1c) and increased respiration rate by inducing a non-phosphorylating energy wasting in the mitochondria [94,95].

By oxidizing rat brain mitochondria with iron ascorbate, Sen et al. [92] found that increased lipid peroxidation is correlated with decreased levels of cardiolipin. In 2011, Liu et al. [34] found that oxidation of tetralinoleoyl cardiolipin (L_4_CL), i.e., the predominant form of cardiolipin in mammals [92,93,96], by cytochrome c and H_2_O_2_ leads to 4-HNE generation (Figure 2c). 4-HNE generation via oxidation of cardiolipin exerts some pathological significance. For example, generation of 4-HNE in vivo has been implicated in the occurrence of atherosclerosis [17]. 4-HNE alters multiple essential functions of brain mitochondria, which plays a pivotal role in the initiation and progression of neurodegenerative diseases [97]. However, the detailed mechanism how 4-HNE damages cells and organs for the occurrence of lifestyle-related diseases has been unknown.

## 5. Beneficial and Detrimental Role of 4-HNE in Various Tissues

Although ubiquitously expressed in the body, GPR40 expression is the most abundant in the brain, and then in the pancreas [49]. GPR40 is activated by medium- and long-chain saturated and unsaturated fatty acids, including ω-6 PUFA [50], and depending on their concentration and extent of oxidation, GPR40 shows dual functions: beneficial or detrimental. For instance, as mentioned above, POMC neurons, associated with both leptin and GPR40 receptors, inhibit food intake, increase energy expenditure, and promote body weight loss via the signals not only from the leptin in the adipose tissue but also from free (non-esterified) fatty acids in the blood. GPR40 activation by DHA protects neurons against the adverse effects of neuroinflammation and insulin resistance in the brain (Figure 2a) [98]. Although signaling through GPR40 decreases in high-fat-diet mice showing cognitive deficits, activation of GPR40 by DHA or by its synthetic agonist, GW9508 (Figure 3b), upregulated c-fos or improved cognitive functions [74,99]. In neurons, the effect of GPR40 activation appears to depend upon the type and concentration of the stimulating fatty acids. For example, in a human neuroblastoma model (SK-N-MC), palmitic acid enhances amyloid β production via GPR40-mediated pathways through mTOR/p70S6K1-mediated HIF-1α expression and NF-κB activation [100]. In the hypothalamic neuronal cell line N43/5, palmitic acid decreases autophagic influx and insulin sensitivity and induces insulin resistance via the excessive activation of GPR40 [101]. Via the sustained stimulation of the GPR40 receptor in the living animals, POMC neurons undergo abnormal cell degeneration/death [20,48,71,72]. Therefore, GPR40 is nowadays becoming a potential research target in both health and diseases of the brain and pancreas.

Similar to GPR40, other closely related G-protein-coupled receptors like GPR109A and GPR120 (also called FFAR4), have also been shown to bind medium- and long-chain fatty acids (e.g., ω-3 DHA and EPA) and produce beneficial effects in diabetes and obesity [102]. Activation of GPR40 regulates insulin secretion and stimulates insulin signaling in pancreatic β-cells, but its role in the brain has not been clear [27,50,103,104,105,106]. The beneficial role of DHA-GPR40-CREB signaling for adult neurogenesis was first suggested by Yamashima and his colleagues [52,54,55,107]. On the contrary, in response to 4-HNE, a metabolic sensor and HCA(2) receptor GPR109A was demonstrated to induce excessive Ca^2+^ mobilization and subsequent cell death in the retinal pigmented and colon epithelial cells [108]. Western blotting analysis of pancreatic tissue after consecutive injections of synthetic 4-HNE in monkeys showed overexpression of the 4-HNE receptor GPR109A with the resultant increases of μ-calpain activation and Hsp70.1 cleavage [7]. In addition, Seike et al. [109] confirmed expression of GPR120 in both the human liver with NASH and the monkey liver after synthetic 4-HNE injections. Using a cultured hepatocyte HepG2 exposed to 4-HNE, they demonstrated that effects of 4-HNE are regulated by activated µ-calpain via GPR120 [109]. Thus, signaling through at least three fatty-acid- and 4-HNE-sensitive G-protein-coupled receptors has important physiological and pathological consequences.

## 6. Injury of Neurons, Hepatocytes, and β-Cells in 4-HNE-Injected Monkeys

Deep-fried foods cooked in vegetable oils such as soybean, rapeseed (canola), sunflower, and corn oils contain abundant 4-HNE generated via heat-induced peroxidation of linoleic acid in the oils [5]. Therefore, after the consumption of deep-fried foods, the concentration of exogenous 4-HNE in the plasma increases rapidly within minutes to hours [110]. As discussed earlier, intake of saturated fatty acids, especially palmitate, leads to the generation of endogenous 4-HNE from cardiolipin (Figure 2b,c). This exogenous and endogenous 4-HNE synergically causes an increase of the 4-HNE levels in the blood and organs with aging. For example, under the age of 40, serum 4-HNE levels are below 0.075 μmol/L, whereas in people aged 60 years old and older, they increase to 0.09 to 0.125 μmol/L even in healthy subjects [111]. Importantly, 4-HNE levels are much higher in those with impaired aldehyde dehydrogenase 2 (ALDH2). ALDH2 is involved not only in the metabolism of acetaldehyde generated by alcohol consumption [112], but also in the metabolism of 4-HNE [113]. Therefore, a reduction in or loss of ALDH2 enzyme activity due to ALDH2 gene mutation causes an increase in 4-HNE level [114]. Subjects with ALDH2*2 (Glu504Lys), a common variant found in about 600 million people of East Asian descent, cannot eliminate toxic aldehydes such as 4-HNE that are associated with a variety of human pathologies [114]. For example, ALDH2 mutation is a risk factor for Alzheimer’s disease [113,114,115,116], and high serum 4-HNE levels are also a major risk factor for type 2 diabetes [45] and NASH [88]. Although available epidemiological data suggest the implication of 4-HNE in these representative lifestyle-related diseases, previous researchers failed to explain the underlying mechanism precisely.

With regard to the implication of 4-HNE in the progression of Alzheimer’s disease, type 2 diabetes, and NASH, the cellular and molecular mechanisms of 4-HNE-induced organ injury should be elucidated in detail. However, detailed analyses, especially focusing on primates, were not carried out until recently. Using Japanese macaque monkeys (*Macaca fuscata*) with a gene homology of ~94% to humans, the author’s group studied the adverse effects of 4-HNE on the primate organs [5]. To replicate blood concentrations of 4-HNE in the human 60s, intravenous injections of 5 mg/week of synthetic 4-HNE (Cayman Chemical, Ann Arbor, MI, USA) were given for 24 weeks (total dose, 120 mg), using very young monkeys with a body weight of 5–7 kg [7,48,109,117]. Six months after the initial injection, the brain, liver, and pancreatic tissues were collected and examined. In the brain, the arcuate nucleus of the hypothalamus, CA1 of the hippocampus, and the precuneus in the parietal lobe were excised, because the former is related to obesity while the latter two are closely related to the occurrence of Alzheimer’s disease. These tissues were used for the histological, immunofluorescence histochemical, electron microscopic, and Western blotting analyses. The Japanese monkeys were utilized because (1) each organ is large enough to achieve all analyses simultaneously in the given monkey, (2) anti-human antibodies can be utilized for the immunohistochemical and Western blotting analyses, and (3) repeated blood sampling and liver biopsy were possible to trace the disease progression.

Upon hematoxylin–eosin staining, many neurons in the arcuate nucleus after the synthetic 4-HNE injections showed necrotic cell death with dissolution of the cytoplasmic organelles and nuclear chromatin. There was no evidence of apoptotic cell death such as nuclear chromatin condensation (apoptotic bodies) or membrane blebbing (Figure 5a, dot circles). Immunofluorescence histochemical analysis demonstrated that the POMC neurons significantly decreased in number (Figure 5d) after the 4-HNE injections (Figure 5c) as compared to the control (Figure 5b). Electron microscopic analysis identified a large number of round lysosomes, measuring 300–500 nm within the neurons of the control monkeys (Figure 5e, circle). In contrast, after the 4-HNE injections, the number of intact lysosomes remarkably decreased, whereas autophagosomes measuring 350–800 nm with an irregular conformation increased (Figure 5f, arrows). These autophagosomes showed a distinct feature from the lysosomes that survived (Figure 5f, open arrow). Furthermore, microcystic changes in the perineuronal dendrites were observed (Figure 5f, asterisks). A decrease in the synaptic vesicles with depositions of the lamellar structure was also seen in the synapses (Figure 5f, circle). The ultrastructure of the POMC neurons after the 4-HNE injections identified lysosomal membrane (Figure 4i, arrows) and mitochondrial membrane damages (Figure 4i, m) [48], as seen in the rodents fed high-fat diets [72]. Similar ultrastructural changes were observed also in the neurons of the hippocampal CA1 and precuneus neurons; there was a substantial decrease in the number of lysosomes and an increase in the number of autophagosomes. These were similar to the ultrastructural changes seen in the rodents fed high-fat diets [72].

During the 4-HNE injections, increased blood levels of AST, ALT, and γ-GTP were confirmed (Figure 6e, closed circles) as compared to the control phase prior to the injections (Figure 6e, open circles). Since increases in AST, ALT, and γ-GTP occurred the week after the first 4-HNE injection (Figure 6e, arrows), this suggested that the exogenous 4-HNE caused acute hepatocyte injury. The surface of the control liver looked reddish-brown (Figure 6d), whereas the liver of the 4-HNE-treated monkeys showed heterogenous, whitish-yellow discoloring (Figure 6f) being intermingled with the dark-brown area. The petechial hemorrhage was seen in the left lobe (Figure 6f, arrow).

Electron microscopic analysis of the control hepatocytes showed membrane-bound, electron-dense lysosomes (Figure 6a, arrow) [117]. In contrast, most of the hepatocytes of the 4-HNE-injected groups showed a remarkable decrease in membrane-bound lysosomes. Compared to the control, the number of autophagosomes increased after the 4-HNE injections (Figure 6b, arrow). Furthermore, in the hepatocytes of the control monkeys, the cytoplasm was filled with mitochondria with normal crista (Figure 6a), whereas after the 4-HNE injections, the mitochondria showed a marked decrease in number and increased disruption and loss of cristae (Figure 6b,c). The complete disruption of the mitochondria resulted in the deposition of the lamellar structure (Figure 6c, arrows). Damaged mitochondria aggregates accumulated in the vicinity of the plasma and nuclear membranes (Figure 6b). The cytoplasm showed a loss of glycogen granules which were present in the control hepatocytes (Figure 6a) and increase in coarsely granular, electron-thin debris (Figure 6b,c).

In mice fed a high-fat, high-sucrose diet (HFHSD), there was a decrease in mitochondrial content, like in the 4HNE-injected monkeys. Furthermore, the relative quantification of mitochondrial membrane phosphatidylcholine showed a 30% reduction compared to the mice fed the standard diet (Figure 1c) [16]. Not only phosphatidylcholine, but also phosphatidylethanolamine, phosphatidylinositol, and phosphatidylserine significantly decreased. Interestingly, the HFHSD mice showed a marked deposition of lipid droplets and decrease in mitochondria in the liver (Figure 1b). In the 4-HNE-treated monkeys, a lipid droplet deposition was observed, but it was much less severe, presumably because they were fed normal chow. However, the concentration of phosphatidylcholine in the monkey liver tissues also showed a reduction of approximately 9.5% [117]. The link between choline/phosphatidylcholine deficiency and hepatic steatosis was well recognized more than half a century ago [118]. Phosphatidylcholine is required for the assembly/secretion of lipoproteins in the liver, for solubilizing cholesterol in bile, and finally, for the efflux of very low-density lipoprotein (VLDL) [119,120,121,122]. The 4-HNE-treated monkeys showed similar features with the HFHSD mice in the reduction of phosphatidylcholine, deposition of fat droplets, and severe damage of mitochondria. It is likely that the HFHS diet in mice induced production of ROS in the mitochondria (Figure 1a), oxidation of cardiolipin involved in the inner membrane (Figure 1c), and generation of 4-HNE (Figure 2b,c).

Betaine-homocysteine S-methyltransferase (BHMT: EC2.1.1.5) is an enzyme that is predominantly found in the liver as a major regulator of choline metabolism [123,124]. BHMT deficiency leads to elevated betaine and homocysteine concentrations associated with the reduction in choline concentration [125]. Consequently, choline deficiency influences hepatic lipid accumulation by reducing the phosphatidylcholine concentration. For example, mice with the gene deletion encoding BHMT showed an approximately 27% decrease in phosphatidylcholine concentrations. They developed fatty livers at 5 weeks after birth, which was due to a decrease in the secretion of VLDL [125]. The increased biosynthesis of phosphatidylcholine stimulates the production of VLDL particles, while the acyl-chains of phosphatidylcholine can modulate VLDL secretion [126]

By the proteomics and Western blot analyses, the monkeys after the 4-HNE injections showed BHMT disorders such as an increase in its carbonylation, an increase in its cleavage, and a decrease in its naïve protein [117]. The carbonylation of BHMT was previously reported using mass spectrometry in a rat model of alcoholic steatosis, which was characterized by the accumulation of fat in the liver 3 and 6 weeks after ethanol exposure [127]. Interestingly, BHMT carbonylation occurs not only via ethanol-derived acetaldehyde but also via ω-6 PUFA-derived 4-HNE, because ‘-aldehyde’ and ‘-nal’ mean the same chemical structure of ‘-CHO’. It is probable that, similar to the case of Hsp70.1, calpain-mediated cleavage of carbonylated BHMT had occurred in the 4-HNE-treated monkeys [117].

Western blotting analysis of the pancreatic tissue after the consecutive injections of synthetic 4-HNE in the monkeys showed an increase in μ-calpain activation, Hsp70.1 cleavage, and overexpression of the 4-HNE receptor GPR109A [7]. In addition, Seike et al. [112] confirmed expression of GPR120 in both the human liver with NASH and the monkey liver after synthetic 4-HNE injections. Using cultured hepatocyte HepG2 exposed to 4-HNE, they demonstrated that the effects of 4-HNE are regulated by activated µ-calpain via GPR120 [112]. Although not so severe as the liver injury, the 4-HNE-treated monkeys showed an interesting cell injury of the pancreas [7]. The pancreas looked normal at the gross inspection, but the Langerhans islet cells after the 4-HNE injections microscopically showed the formation of many tiny vacuoles (Figure 7a, circle) as compared to the control. These vacuoles were thought to be enlarged rough ERs (Figure 7b,c, stars). A small number of nuclei showed dissolution of chromatin or punctuate condensation, but neither apoptotic bodies nor membrane blebbings were observed on both light and electron microscopic observations [7]. Electron microscopy of the β-cells was characterized by insulin secretory granules which had an electron-opaque core of 300–400 nm with a clear halo (Figure 7b, β, arrows). The δ-cells exhibit neuron- or trumpet-like morphology with cytoplasmic processes extending from the islet capillaries (Figure 7b,c, δ). After 4-HNE treatment, there was a marked decrease in insulin granules (Figure 7b,c, arrows) and somatostatin (Figure 7b,c, white arrows) granules in the Langerhans cells compared to the control (Figure 7d, arrows). Both the β- and δ-cells had an increased number of vacuoles, possibly enlarged rough ERs (Figure 7c, stars). Finally, the number of autophagosomes containing degenerating mitochondria or mitochondria-derived debris increased in the β-cells from the 4-HNE injected monkeys, while the number of intact lysosomes markedly decreased (Figure 7c, circles).

A 30 kDa cleaved form of heat-shock protein 70.1 (Hsp70.1) from the 4-HNE-treated monkeys in the liver and pancreatic tissues increased relative to those of the control-treated monkeys (Figure 8a). An anti-μ-calpain antibody that reacts only with the activated form [128,129] identified activated μ-calpain immunoreactivity of small granules in Kupffer cells in the control hepatocytes (Figure 8c), and after 4-HNE treatment, the immunoreactivity of both Kupffer cells and hepatocytes remarkably increased (Figure 8d). Minimal co-staining of Hsp70.1 with activated μ-calpain was observed only in the Kupffer cells of livers from the control-treated group (Figure 8e), whereas after 4-HNE treatment, Hsp70.1 co-localization with activated μ-calpain greatly increased in the Kupffer cells (Figure 8f, arrows) and in hepatocytes (Figure 8f, circle). The common ultrastructural changes induced by synthetic 4-HNE among neurons (Figure 5f), hepatocytes (Figure 6b), and β-cells (Figure 7b), were severe degeneration and loss of mitochondria, a decrease in intact lysosomes, and an alternative increase in autophagosomes. Although 4-HNE was reported to activate caspase-3 and release cytochrome c that cause apoptosis [130,131], 4-HNE-injected monkeys did not show any evidence of apoptotic cell death in the brain, liver, and pancreas upon examination via both light and electron microscopy (Figure 5, Figure 6 and Figure 7).

## 7. Carbonylation and Cleavage of Hsp70.1 Cause Diverse Cell Death

Hsp70.1 (also called Hsp70, Hsp72 in humans) is the most conserved molecular chaperone which is crucial for stabilizing the lysosomal limiting membrane against various stresses [132,133]. Hsp70.1 is induced in response to many forms of brain injury such as stroke, trauma, status epilepticus, etc., and its overexpression plays a protective role against neuronal ischemic injury [134,135]. After transient brain ischemia in monkeys, for example, hippocampal CA1 neurons developed delayed neuronal death on day 5 [128,129]. Proteomics analysis of CA1 neurons of these ischemic monkeys showed a marked increase in Hsp70.1 levels as compared to the non-ischemic controls. Two-dimensional gel electrophoresis with immunoblot detection of carbonylated protein analysis (2D Oxyblot) of the hippocampal tissues after ischemic insult showed about a 9-fold increase on day 3 and 4-fold increase on day 5 in carbonylated Hsp70.1 [136], consistent with the data of Sultana et al. [137] obtained from patients with mild cognitive impairment (MCI) and early Alzheimer’s disease. Intriguingly, in the postischemic monkeys, the matrix-assisted laser desorption ionization time-of-flight/time-of-flight analysis showed a decrease in molecular weight at the key site Arg469 from 157.20 to 113.12. Therefore, the specific oxidative injury ‘carbonylation’ occurred at Arg469 of Hsp70.1 [136,138]. It is likely that carbonylation of the key site Arg469 increases vulnerability of Hsp70.1 to calpain-mediated proteolysis (Figure 8a,b) [8,139,140].

Previously, stress signaling proteins corresponding to the heat shock response such as Hsp70.1 and Hsp90, as well as proteins involved in redox regulation such as glutathione-S-transferase Pi, were identified as being modified by 4-HNE in vivo [140,141]. Using normal hippocampal CA1 tissues resected from young, healthy monkeys, calpain-mediated cleavage of the carbonylated Hsp70.1 was demonstrated to occur in vitro during incubation with 4-HNE or H_2_O_2_ (Figure 8b) [142]. Subsequently, this proteolysis was confirmed using diverse brain tissues such as the thalamus, putamen, and medulla oblongata [143]. Importantly, incubation of the CA1 tissue with 1.0 mM H_2_O_2_ induced the same extent of calpain-mediated cleavage of the oxidized Hsp70.1 as 500 µM 4-HNE (Figure 8b). Generation of endogenous 4-HNE and the subsequent carbonylation of the tissue Hsp70.1 were completed within 2 h of H_2_O_2_ treatment. It was also found that endogenously generated 4-HNE by H_2_O_2_ treatment (Figure 8b, 1.0 mM H_2_O_2_) exerted the same effect as exogenously applied 4-HNE (Figure 8b, 500 µM 4-HNE) on Hsp70.1 carbonylation. Despite the unique anatomical, physiological, and biochemical characteristics of each organ in different diseases, a common cascade of cell degeneration/death which is caused by 4-HNE appears likely [5,8,139]. If this is the case, how does carbonylation of Hsp70.1 lead to necrotic cell death? Is there another player that is activated by cell stress and can influence carbonylated Hsp70.1?

Calpain (EC 3.4.22.17) is an intracellular, non-lysosomal, Ca^2+^-dependent, papain-like protease. Calpain comprises two isoforms, μ-calpain (also called calpain-1) and m-calpain (also called calpain-2), reflecting the μM and mM levels of Ca^2+^ required for activation, respectively. There are abundant data on the implication of μ-calpain in the pathogenesis of Alzheimer’s disease [144]. For example, Taniguchi et al. [145] found that μ-calpain is activated more than 7-fold in the brains of patients with Alzheimer’s disease, as compared to age-matched brains of healthy individuals. Acute, extensive ischemia during stroke and long-standing mild ischemia due to arteriosclerosis associated with aging are both associated with increased activated µ-calpain relative to healthy elderly people. In addition, 4-HNE (50 to 400 μM) increased Ca^2+^ levels in a dose-dependent manner [146] and causes translocation of µ-calpain to membranes, indicating that 4-HNE itself can activate µ-calpain (Figure 8c,d), as shown by Seike et al. [109]. Interestingly, an increase in activated µ-calpain (Figure 8c,d) and its co-localization with oxidized Hsp70.1 in 4-HNE-treated monkeys were found relative to the control (Figure 8e,f) [109,117]. Since carbonylated Hsp70.1 is vulnerable to activated µ-calpain, Hsp70.1 carbonylation may trigger its proteolysis [8].

Except for carbonylated Hsp70.1 and BHMT, activated µ-calpain cleaves other lysosomal membrane proteins such as Lamp-2 [147,148] and subunit b2 of v-ATPase [149]. This also facilitates lysosomal membrane permeabilization/rupture causing the release of lysosomal cathepsin enzymes into the cytosol and the degradation of cell constituents. The release of lysosomal cathepsin B, D, etc., can depolarize mitochondrial membranes and induce further ROS production [150]. This ‘calpain–cathepsin hypothesis’ (Figure 9, dot rectangle) [5,129,138,139,151] suggests a role for µ-calpain and cathepsin in the occurrence of lysosome-induced cell death.

## 8. Common Molecular Cascade Between Alzheimer’s Disease and Type 2 Diabetes

The concept of the ‘calpain-cathepsin hypothesis’ (Figure 9) is essentially in line with the molecular mechanism of autophagy-lysosomal-associated neuronal death in neuro-degenerative diseases [154]. If considering implication of the autophagy-lysosomal failure in the pancreas pathology [5,7], the common molecular cascade may be present between neurodegenerative diseases and lifestyle-related diseases, especially between Alzheimer’s disease and type 2 diabetes. A similarity between the pathologies associated with Alzheimer’s disease and type 2 diabetes was first postulated by Hoyer in 2002, because both diseases exhibit declines in glucose uptake, insulin levels, insulin binding, and tyrosine kinase activity [155]. The impairment of glucose metabolism is interconnected with the core pathophysiology of Alzheimer’s disease. Insulin resistance that defines type 2 diabetes contributes not only to hyperglycemia but also to hyperlipidemia, inflammation, oxidative stress, etc. Hyperglycemia is one of the major sources of ROS and leads to modulation of various metabolic downregulatory pathways (Figure 1a). For example, increased glucose causes glucose oxidation and production of hydrogen peroxide and hydroxylradical [14]. Insulin resistance in the brain was defined as the inability of neurons or glia cells to respond to insulin action, resulting in impairments in the synaptic, metabolic, and immune response functions [156]. Type 2 diabetes is associated with brain insulin resistance, a feature of Alzheimer’s disease; however, the mutual relation of the two diseases with aging remains unclear.

There is a strong correlation between chronic intake of a high-fat diet and the development of neuroinflammation, especially in the hypothalamus [27]. Palmitate intake induces mitochondrial oxidative stress via the generation of ROS during its β-oxidation [157] (Figure 1 and Figure 2) [158,159,160]. Consumption of a high-fat diet reduces the mitochondrial fusion protein mitofusin 2 (MFN2) in hypothalamic neurons, and this results in loss of mitochondrial–ER contact, occurrence of ER stress, and the development of ER-stress-induced leptin resistance [161]. In the animal model, loss of mitochondrial–ER contacts also reduces mitochondrial metabolism and the cellular redox state [162,163]. These observations were corroborated in C57BL/6 mice fed high-fat diets, showing a decrease in MFN2 expression in the arcuate nucleus (Figure 2a) [164]. In addition, since high-fat diets reduced the mitochondrial-dependent Ca^2+^ uptake capacity, the resultant decrease in hypothalamic neuronal excitability caused impaired energy control in the hypothalamus during obesity [165].

Since IL-1β is a key mediator in the neuroinflammatory pathway, the complex crosstalk between calpain and NLRP3 activations [166] leads to insulin resistance in the brain. NLRP3 inflammasome is a molecular platform mediating the maturation of IL-1β in response to stress conditions. To produce mature IL-1β, two distinct signals are indispensable. The priming step is the activation of NF-κB to initiate the transcription of pro-IL-1β and NLRP3, which is expressed at low levels under resting conditions [167]. The second activation step is the assembly of the NLRP3 inflammasome to enable processing and release of IL-1β [25]. Interplay between brain insulin resistance, neuroinflammation, and peripheral metabolic dysregulation is likely the linkage between type 2 diabetes and Alzheimer’s disease [153,166,167,168].

ROS trigger µ-calpain activation, which contributes to amyloid β42 production, CDK5-mediated tau phosphorylation, NLRP3 inflammasome activation, IL-1β processing, and lysosomal membrane rupture/permeabilization [139,151,152,153,166,168]. At the cell membrane, amyloid β42/amylin oligomer accumulation is correlated with the activation of the NLRP3 inflammasome. In type 2 diabetic patients, an overload of free fatty acids also activates the NLRP3 inflammasome [23]. Cystatin C may be a further link between Alzheimer’s and type 2 diabetes, since polymorphisms were found in both conditions, and cystatin C binds amyloid β and fosters aggregation of amyloid β40 and amyloid β42 at certain concentrations [169].

In summary, after intake of high-fat diets, β-oxidation of free fatty acids in the mitochondria produces ROS which enhance oxidation of linoleic acids within biomembranes, especially in the mitochondrial inner membranes to increase endogenously generated 4HNE (Figure 2) [27]. In addition, 4-HNE is generated during heating vegetable oils containing linoleic acids, which is incorporated into the body via deep-fried foods. Since 4HNE is amphiphilic, this endogenous and exogenous 4-HNE diffuses within and outside the cells and reacts with targets like senile and atheromatous plaques in the brain or arterial wall. 4-HNE can increase lysosomal injuries by facilitating activated μ-calpain-mediated cleavage of the carbonylated Hsp70.1 (Figure 9).

It is conceivable that Alzheimer’s disease, type 2 diabetes, and NASH are parallel pathological phenomena induced by 4-HNE accumulation (Figure 9). Tissue and serum levels of 4-HNE depend not only on its generation but also on how it is metabolized and eliminated [29,36,42,113,114]. ALDH2, glutathione S-transferase, and aldose reductase are major candidate enzymes that metabolize and regulate 4-HNE levels. It is likely that detoxifying 4-HNE via activated ALDH2 and/or protecting Hsp70.1 from oxidative injury would be an effective strategy for the prevention and treatment of these lifestyle-related diseases.

## 9. Conclusions

The ‘calpain–cathepsin hypothesis’ formulated by the author postulates calpain-mediated damage of lysosomal limiting membranes and subsequent cathepsin release, which represent a central cascade for both ischemic and degenerative cell death. Mitochondria and lysosomes are particularly vulnerable to 4-HNE-induced damage. Increases in both endogenous and exogenous 4-HNE, combined with age-dependent ischemia, may overactivate µ-calpain, which can cleave the lysosomal stabilizer protein, Hsp70.1, especially after it is carbonylation by ROS, and thus induce lysosome-mediated cell death via the cathepsin leakage. The high abundance of mitochondria in the brain, pancreatic, and liver tissues and their contribution to ROS generation during pathological stress put these organs at risk of 4-HNE-induced toxicity.

## Figures and Tables

**Figure 3 nutrients-16-04171-f003:**
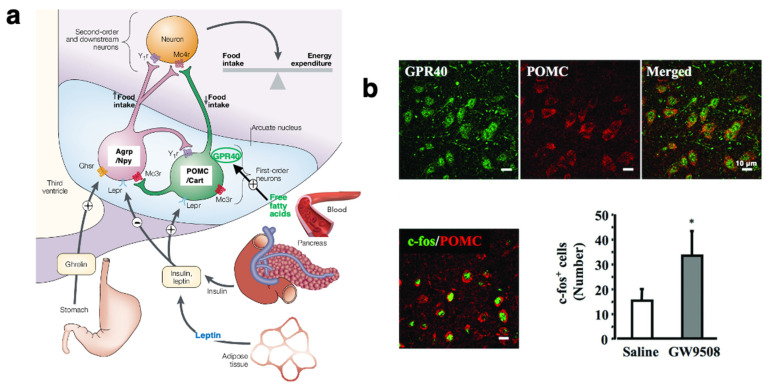
POMC neurons of the arcuate nucleus of the hypothalamus and GPR40. (**a**) There are two populations of centrally projecting neurons in the arcuate nucleus (light blue area) which respond to diverse neuropeptides and free fatty acids. Agrp (agouti-related protein) and Npy (neuropeptide Y) stimulate food intake and decrease energy expenditure in Agrp/Npy neurons. In contrast, POMC (pro-opiomelanocortin) and Cart (cocaine- and amphetamine-regulated transcript) inhibit food intake and increase energy expenditure in POMC/Cart (POMC) neurons. So, POMC neuronal degeneration/death play a crucial role in the development of hyperphagia and perpetuation of obesity. (Adapted with permission from Ref. [73], Barsh and Schwartz, 2002.) (**b**) As POMC neurons express GPR40, administration of its agonist, GW9508, activates POMC neurons in the arcuate nucleus with the induction of the proto-oncogene c-fos. Presumably, similar GPR40 activation occurs in POMC neurons in response to free fatty acids in the blood. bar = 10 μm. * *p* < 0.05 (Adapted with permission from Ref. [74], Nakamoto et al., 2013).

**Figure 4 nutrients-16-04171-f004:**
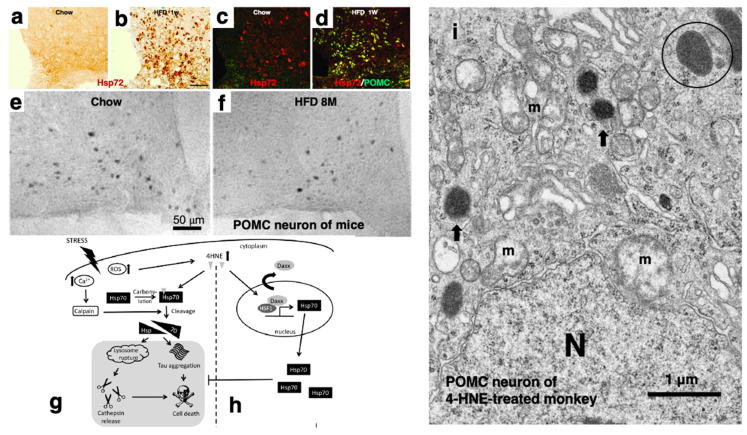
POMC neuronal injury in rodents fed high-fat diets (HFDs) (**a**–**f**) and in the monkey after 4-HNE injections (**i**). (**a**–**d**) Immunohistochemical analysis of Hsp72 (compatible with the Hsp70.1 of primates) in rats fed a non-purified diet (Chow) (**a**,**c**) or high-fat diet (HFD) (**b**,**d**) for 7 days (1 W). A remarkable upregulation of Hsp72 is induced by HFD. Immunofluorescence shows colocalization of Hsp72 (red) with POMC peptide (green) (**c**,**d**). Double staining (merged yellow color) indicates Hsp72 upregulation at the POMC neurons by the HFD. (**e**,**f**): Representative images of POMC neurons in the mice fed either Chow (**e**) or HFDs (**f**) for 8 months (8 M). Compared with Chow mice (**e**), HFD (**f**) mice show a significant loss of POMC neurons. bar = 50 μm. POMC, proopiomelanocortin. (**a**–**f**: adapted with permission from Ref. [72]; Thaler et al., 2012) (**g**,**h**) 4-HNE is a typical Janus-faced molecule, with both cell-toxic (**g**) and protective (**h**) effects. Enhanced Hsp70 expression levels might counteract lysosomal rupture and tau aggregation. HNE induces nuclear export of the HSF1 inhibitor Daxx, which results in HSF1 activation to induce Hsp70 upregulation. Daxx, inhibitor death associated protein 6. (**g**,**h**: adapted with permission from Ref. [75]; Penke et al., 2018.) (**i**) Electron microphotograph of the 4-HNE-treated monkey showing lysosomal membrane permeabilization in lysosomes (arrows) which shows a marked contrast with the membrane-bound lysosome (circle). Disruption of the mitochondrial inner membranes (m) is also appreciated. N: nucleus, bar = 1 μm. (**i**: Adapted with permission from Ref. [48]; Yamashima et al., 2022).

**Figure 5 nutrients-16-04171-f005:**
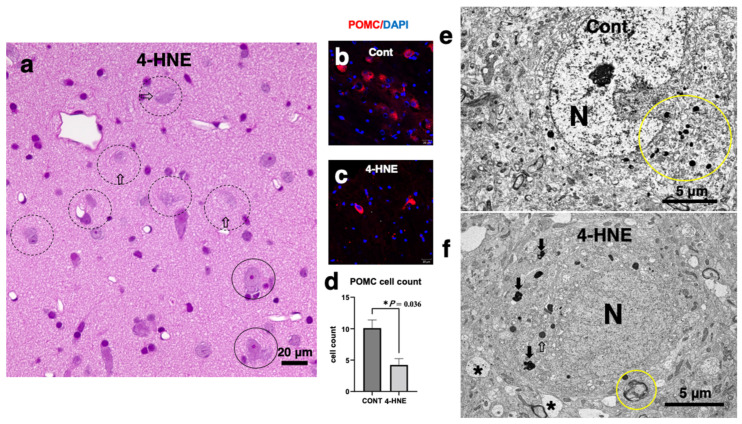
Light and electron microphotographs of POMC neurons after injections of 5 mg/week of 4-HNE for 24 weeks. (**a**) Neurons of the arcuate nucleus show degeneration (circles) or cell death (dot circles) with dissolution of the nuclear chromatin and cytoplasm (open arrows). Apoptotic bodies were not observed. Note tiny vacuolations around the neuron (open arrows) which are thought to be enlarged synapses (**f**, asterisks). (Hematoxylin–eosin staining; bar = 20 μm.) (**b**–**d**) Based on the immunofluorescence histochemical analysis using the anti-POMC antibody, the number of POMC neurons significantly decreased (**d**) in the monkeys after the 4-HNE injections (**c**) compared to the control (**b**). (**a**–**d**: Adapted with permission from Ref. [48]; Yamashima et al., 2022.) (**e**,**f**) Via the electron microscopic analysis, a large number of round or oval lysosomes measuring 300–500 nm in diameter (**e**: circle) were seen within the neurons of the healthy monkeys. In contrast, after the 4-HNE injections, the number of round or oval vivid lysosomes (**f**, open arrow) remarkably decreased, whereas the number of autophagosomes measuring 350–800 nm with an irregular configuration increased (**f**: arrows). 4-HNE also induced synaptic degeneration, forming microcysts (**f**, asterisks) or a lamellar structure (**f**, circle). (**e**,**f**): Bar = 5 μm.

**Figure 6 nutrients-16-04171-f006:**
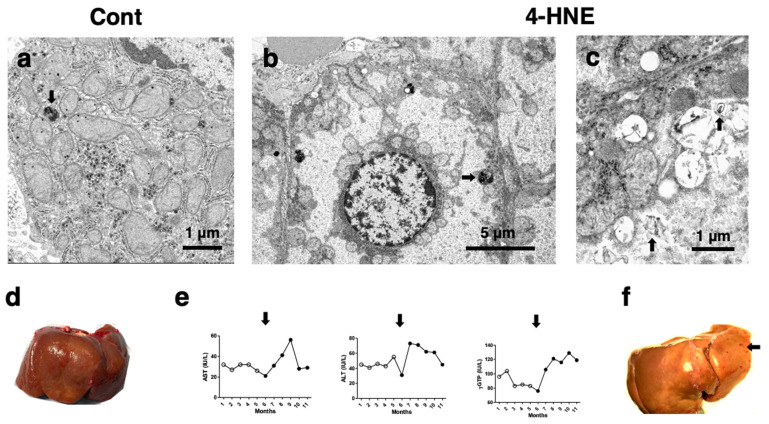
Electron microphotographs of hepatocytes (**a**–**c**), blood analysis data (**e**), and gross inspection of the liver before (**d**) and after (**f**) 4-HNE injections in young, healthy monkeys. (**a**–**c**) The control hepatocytes ultrastructurally showed membrane-bound, electron-dense lysosomes (**a**, arrow, bar = 1 μm), whereas the hepatocytes after the 4-HNE injections showed a remarkable decrease in intact lysosomes and an increase in autophagosomes (**b**, arrow, bar = 5 μm). The control hepatocyte contained numerous mitochondria (**a**), but after the 4-HNE injections, the mitochondria showed a marked decrease, with loss of cristae, and the cytoplasm contained abundant granular debris (**b**). Degeneration of the mitochondrial inner membranes may lead to depositions of the lamellar structure (**c**, arrows, bar = 1 μm). (**d**,**f**) The control liver looked reddish-brown (**d**), whereas the 4-HNE-treated liver showed heterogenous, whitish-yellow discoloring (**f**). A petechial hemorrhage was seen in the left lobe (**f**: arrow). (**e**) The blood data during the 4-HNE injections showed increased levels of AST, ALT, and γ-GTP (**e**: closed circles; arrows indicate start of the 4-HNE injection) compared to the control phase (**e**: open circles). (**d**–**f**: Adapted with permission from Ref. [117]; Yamashima et al., 2023b).

**Figure 7 nutrients-16-04171-f007:**
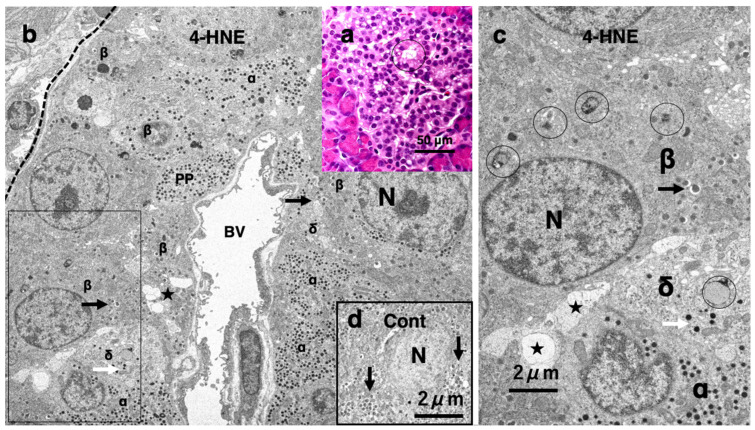
Light and electron microphotographs of the pancreatic Langerhans cells before (**d**) and after (**a**–**c**) 4-HNE injections in young, healthy monkeys. (**a**) Via light microscopic observation, the Langerhans cells show formation of numerous vacuoles (**a**, circle). (Hematoxylin–eosin staining. bar = 50 μm.) (**b**–**d**) Based on electron microscopic observation, the β-cells (**β**) were characterized by insulin secretory granules which had an electron-opaque core of 300–400 nm with a clear halo (**c**, arrows). The δ-cells (**δ**) exhibit neuron- or trumpet-like morphology with cytoplasmic processes extending from the islet capillaries. The most remarkable change in the Langerhans cells after the 4-HNE injections was a remarkable decrease in insulin granules (**b**,**c** arrows) and somatostatin (**b**,**c** white arrows) granules compared to the control (**d**, arrows). Both the β- and δ- cells showed numerous microvacuole formations which were thought to be enlarged rough ERs (**b**,**c**, stars). In the β-cells after the 4-HNE injections, autophagosomes containing degenerating mitochondria or mitochondria-derived debris were seen (**c**, circles). N: nucleus, PP: pancreatic polypeptide cells (PP-cells), **α**: α-cells, BV: blood vessel. (Adapted with permission from Ref. [7]; Boontem and Yamashima, 2021.) (**b**–**d**) Bar = 2 μm. The rectangle in (**b**) was enlarged as (**c**).

**Figure 8 nutrients-16-04171-f008:**
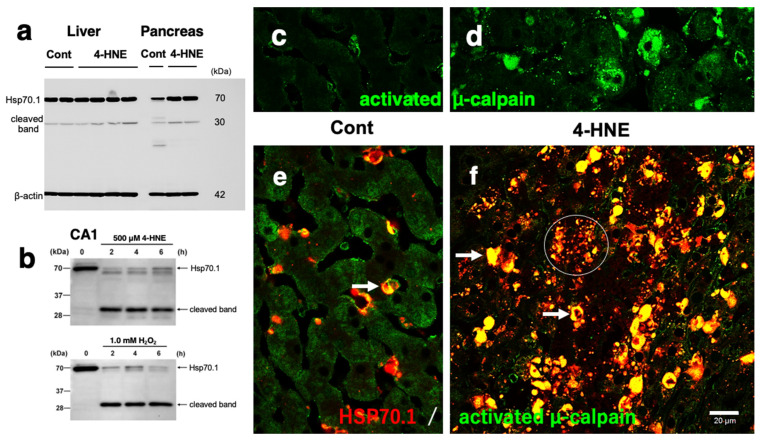
Western blotting (**a**,**b**) of the liver, pancreas, and hippocampal CA1, and immunohistochemical (**c**–**f**) analyses of the liver after 4-HNE injections/incubations. (**a**,**b**) Western blotting; Hsp70.1 was upregulated in both the liver and pancreas after 4-HNE injections, showing an increase in 30 kDa cleaved bands compared to the controls (**a**, Cont). During the incubation of the fresh CA1 tissues in 500 μM 4-HNE or 1.0 mM H_2_O_2_ (**b**), H_2_O_2_ shows the same extent of cleavage as 4-HNE during incubation for 2, 4, and 6 h compared to the control (0 h). This means that free radicals were produced immediately in the tissue by the H_2_O_2_-induced oxidation of biomembranes to generate 4-HNE. (**c**–**f**) Based on the immunofluorescence histochemical staining of the liver, activated μ-calpain immunoreactivity was negligible in the control hepatocytes (**c**), whereas after the 4-HNE injections, the immunoreactivity remarkably increased (**d**). In the control liver, co-staining of Hsp70.1 with activated μ-calpain was observed only in the Kupffer cells (**e**, arrow) being positive for CD68. In contrast, after the 4-HNE injections, a remarkable increase in the merged immunoreactivity of Hsp70.1 and activated μ-calpain was seen in both Kupffer cells (**f**, arrows) and hepatocytes (**f**, circle). (**c**–**f**: Adapted with permission from Ref. [117]; Yamashima et al., 2023b).

**Figure 9 nutrients-16-04171-f009:**
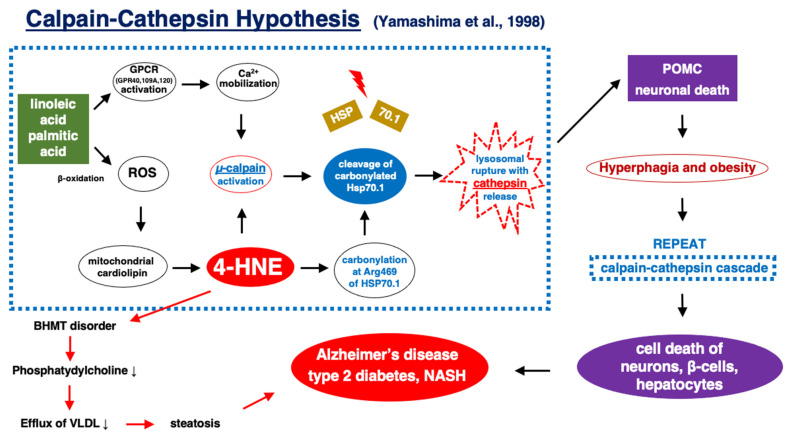
Flowchart from the incorporation of excessive linoleic and palmitic acids to the progression of lifestyle-related diseases. As a common cause of Alzheimer’s disease, type 2 diabetes, and NASH, such a common cascade may proceed in the corresponding cells like ‘excessive fatty acids like linoleic and palmitic acids—activation of GPCR (GPR40 for neurons, GPR109A for β-cells, and GPR120 for hepatocytes, respectively) and the resultant Ca^2+^ mobilization—excessive β-oxidation and production of ROS in mitochondria—generation of 4-HNE via oxidization of mitochondrial cardiolipin—Ca^2+^- and/or 4-HNE-induced µ-calpain activation—Hsp70.1 carbonylation—cleavage of the carbonylated Hsp70.1 by activated µ-calpain—lysosomal membrane disintegrity—cathepsin release—cell degeneration/death’. Along with this cascade, 4-HNE carbonylates BHMT, which leads to the decrease in phosphatidylcholine, impairments of VLDL efflux, and depositions of triglycerides in the liver. Presumably, 4-HNE is the common root substance of Alzheimer’s disease, type 2 diabetes, and NASH. GPCR: G-protein-coupled receptors, ROS: reactive oxygen species, 4-HNE: 4-hydroxy-2-nonenal. (modified from Ref. [129]; Yamashima et al., 1998, Ref. [138]; Yamashima and Oikawa, 2009, Ref [152,153]; Yamashima, 2013, 2016).

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
