# Peer review of "4-Hydroxynonenal from Mitochondrial and Dietary Sources Causes Lysosomal Cell Death for Lifestyle-Related Diseases"

_nutrients, 2024, doi:10.3390/nu16234171_

Round 1
Reviewer 1 Report
Comments and Suggestions for Authors
Peer review report on ‘4-Hydroxynonenal from Mitochondrial and Dietary Source Causes Lysosomal Cell Death for Lifestyle-Related Diseases’.
Manuscript I.D: nutrients-3346429.
This review reports on up-to-date research into the increasing evidence of the role that 4-hydroxynonenal (4-HNE) plays in the development of modern-day diseases such as type 2 diabetes, Alzheimer’s disease, and cardiovascular diseases. There are basically two ways in which 4-HNE is generated in the body; the first is from the ingestion of saturated fatty acids which leads to their β-oxidation in the mitochondria producing ROS, and the subsequent generation of endogenous 4-HNE via the oxidation of mitochondrial cardiolipin, and the second from the direct ingestion of 4-HNE from heat treated ω-6 PUFA in foods resulting in the presence of exogenous 4-HNE. With age, the levels of 4-HNE increase causing inflammatory organ damage and other pathologies. It was proposed that detoxifying the body of 4-HNE in some way could possibly contribute to the decline of these diseases.
The subject matter is very complex, with extensive references. The research is cleverly integrated to make a cohesive whole. It is well-written, and the English is generally good, although it could benefit from the attentions of an English expert just to give a final polish. The diagrams are excellent.
Some comments.
In the title, the word ‘sources’ would be better than ‘source’.
Please add the Figures to the text in the appropriate places. It is very disconcerting having to constantly flip between the text and the Supplementary Material.
Line 276. What is the ‘special condition’?
Line 649. Do you mean ‘status epilepticus’?
Line 679. Please rewrite this sentence.
Line 723. Regarding the impairment of glucose metabolism and these diseases would you mention here that increases in kynurenine pathway metabolites appear to be implicated as well. Is there any correlation between 4-HNE and the kynurenine pathway that you know of?
(Minhas et al, Restoring hippocampal glucose metabolism rescues cognition across Alzheimer’s disease pathologies, Science, 23 Aug 2024, 385 (6711) DOI: 10.1126/science.abm6131
Line 788. Do you mean induced?
Author Response
Nutrients, Special Issue
"Dietary Fats: Beneficial or Detrimental for Lifestyle-Related Diseases"
Nov 27, 2024
Dear Reviewers
This is to deeply grateful for the careful reading and sugesting revisions necessary by two instructive reviewers. By revising the comments of Reviewer 1, I would like to resubmit my review paper, entitled ‘4-Hydroxynonenal from Mitochondrial and Dietary Sources Causes Lysosomal Cell Death for Lifestyle-related Diseases’ to nutrients for the possible publication.
The revisions were thoroughly done according to the comments of Reviewer 1. I shall describe the revised portions as shown below, which, I do hope, would satisfactory, and the reviewers could recommend the final acceptance of the revised manuscript for the Special Issue of nutrients.
With warmest regards,
Tetsumori Yamashima, MD, PhD
Dept. of Psychiatry and Behavioral Science, Kanazawa University Graduate School of Medical Sciences, Takara-machi 13-1, Kanazawa, 920-8041 Japan.
Some comments.
- In the title, the word ‘sources’ would be better than ‘source’.
# ‘source’ was changed to ‘sources’.
- Please add the Figures to the text in the appropriate places. It is very disconcerting having to constantly flip between the text and the Supplementary Material.
# Very Sorry ! The Figures were all added to the appropriate places for you and readers to read without diificulty.
3) Line 276 (278). What is the ‘special condition’?
# The ‘special condition’ was changed to ‘metabolic alterations’.
4) Line 649 (665). Do you mean ‘status epilepticus’?
# Exactly, so! Thank you so much for the careful suggestion. The author is extremely grateful.
5) Line 679 (695). Please rewrite this sentence.
# ‘If this is the case, how carbonylation of Hsp70.1 leads to necrotic cell death?’ was revised as following.
‘If this is the case, how does carbonylation of Hsp70.1 lead to necrotic cell death?’
6) Line 723 (742). Regarding the impairment of glucose metabolism and these diseases would you mention here that increases in kynurenine pathway metabolites appear to be implicated as well. Is there any correlation between 4-HNE and the kynurenine pathway that you know of?
(Minhas et al, Restoring hippocampal glucose metabolism rescues cognition across Alzheimer’s disease pathologies, Science, 23 Aug 2024, 385 (6711) DOI: 10.1126/science.abm6131
# This is an important issue. However, to the best of my knowledge, there are no previous reports concerning the interaction of 4-HNE toindoleamine-2,3-dioxygenase, i.e. the kynurenine pathway.
7) Line 788. Do you mean ‘induced’?
Yes, exactly. It was changed to ‘induced by’.
Reviewer 2 Report
Comments and Suggestions for Authors
The posted manuscript "4-Hydroxynonenal from Mitochondrial and Dietary Source Causes Lysosomal Cell Death for Lifestyle-Related Diseases" has five important supporting comments:
The research extensively examines 4-hydroxynonenal (4-HNE) in oxidative stress-induced cellular damage and links dietary and mitochondrial sources to lifestyle-related disorders. This method illuminates an understudied molecular process.
This publication connects dietary patterns (e.g., ω-6 PUFA consumption) to mitochondrial dysfunction, using experimental evidence from monkey research. This link is essential for understanding and preventing type 2 diabetes, Alzheimer's, and NASH.
Japanese macaque monkeys closely mimic human physiology, making them useful for translation. This supports the paper's findings on 4-HNE-induced organ damage and lifestyle illnesses.
The research examines molecular mechanisms such the calpain-cathepsin cascade, mitochondrial failure, and lysosomal membrane instability. Electron micrographs clarify and reinforce statements.
Novel therapeutic targets, such as decreasing 4-HNE toxicity, could help prevent and treat metabolic and neurological disorders. This translational approach broadens its scientific and medical appeal.
These features make the work a good contender for publication with little changes.
Author Response

(The authors gave the same response as above.)
